# Multiple Cesarean Section Outcomes and Complications: A Retrospective Study in Jazan, Saudi Arabia

**DOI:** 10.3390/healthcare11202799

**Published:** 2023-10-22

**Authors:** Maha Murtada, Nasser Hakami, Mohamed Mahfouz, Amani Abdelmola, Ebtihal Eltyeb, Isameldin Medani, Ghadah Maghfori, Atheer Zakri, Ahlam Hakami, Ahmed Altraifi, Ali Khormi, Uma Chourasia

**Affiliations:** 1Obstetrics and Gynecology Department, Jazan University, Jazan 82621, Saudi Arabia; mmurtada@jazanu.edu.sa (M.M.); iemedani@jazanu.edu.sa.com (I.M.); ahlamhakami@jazanu.edu.sa (A.H.); aaltraifi@jazanu.edu.sa (A.A.); alikhormi@jazanu.edu.sa (A.K.); uchourasia@jazanu.edu.sa (U.C.); 2Surgery Department, Jazan University, Jazan 82621, Saudi Arabia; nhakami@jazanu.edu.sa; 3Family and Community Medicine Department, Jazan University, Jazan 82621, Saudi Arabia; mmahfouz@jazanu.edu.sa (M.M.); aabashar@jazanu.edu.sa (A.A.); 4Pediatrics Department, Jazan University, Jazan 82621, Saudi Arabia; 5Ministry of Health Jazan City, Jazan 45142, Saudi Arabia; 201803528@stu.jazanu.edu.sa.com (G.M.); 201803544@stu.jazanu.edu.sa.com (A.Z.)

**Keywords:** cesarean section, outcome, complications, Saudi Arabia

## Abstract

Background: Given the increase in the rate of cesarean sections (CSs) globally and in Saudi Arabia, this study was conducted to assess the maternal and perinatal complications after repeat cesarean sections in the studied population. Methods: This retrospective study was conducted by reviewing the records of all women who underwent CSs between January and July 2023 in three hospitals in the Jazan region of Saudi Arabia. Results: Of the 268 women studied, 195 (72.7%) had a CS for the first or second time and 73 (27.3%) had two, three, or four previous CSs (repeat CS). The most common maternal intra-operative complications reported by the repeat CS group were intra-peritoneal adhesions (7.5%) and fused abdominal layers (7.1%) while the most common postoperative complications were the need for blood transfusion (22%) and UTIs (3%). The most common neonatal complications were a low Apgar score (19%), needing neonatal resuscitation (2.6%), and intensive care admission. In addition, 3.7% of mothers failed to initiate breastfeeding in the first 24 h. Conclusions: The frequent complications were intra-peritoneal adhesions, fused abdominal wall layers, blood transfusion, and postoperative infections which were overcome by the optimal hospital care. However, the frequent neonatal complications were a low Apgar score, needing neonatal resuscitation, and intensive care admission.

## 1. Introduction

Cesarean section (CS) is a common obstetric procedure used to overcome problems associated with vaginal delivery, such as cephalo-pelvic disproportion and fetal distress. However, it carries maternal and fetal risks. The maternal risks include infection, anesthetic complications, surgical injury, bleeding, and thromboembolism [1]. Repeat CSs also increase the risk of dense adhesions, bladder injury, bowel injury, and incision-related problems like wound dehiscence. Both maternal and fetal complications are expected to increase in emergency operations compared to elective ones [2].

According to information from the World Health Organization (WHO), the rate of CSs has significantly risen globally in the last thirty years [3]. Despite increasing CS rates, the maternal mortality associated with it is decreasing due to improved anesthetic techniques, availability of antimicrobial agents, and modern blood banking techniques [4]. The risk of intra-operative complications and uterine rupture is increased in women who have had repeat CSs, making these patients a high-risk group [5]. There is concern about the increased CS rate because of the associated elevated morbidity and mortality compared to births through the vaginal route. There is a significant increase in severe maternal morbidities, such as postpartum hemorrhage, admission to intensive care units, and a hospital stay of more than seven days. Perinatal complications include a low Apgar score and significant increase in the length of stay in the NICU (more than seven days) [6].

In Saudi Arabia, cesarean delivery (CD) is one of the most commonly performed surgical procedures. A recent study in 2022 showed a significant increase in CD rates attributed to clinician practice rather than maternal factors [7]. Although the WHO recommends CD rates to be between 10 and 15%, Saudi Arabia’s CD rates have reached 25% [3,8]. Furthermore, in one study involving 14 administrative regions of Saudi Arabia, the CD rate was noticed to have increased by 80.2% in ten years, with significantly increased rates in the kingdom’s northern region [9].

Growing evidence has shown that having multiple CSs can lead to more health problems for mothers. Women who have had one previous CS should be considered at a higher risk and allowed to have a vaginal birth if it is a safe option. The profile of obstetric CSs has been investigated in different parts of KSA [8]; however, no study has been conducted in Jazan in southwest Saudi Arabia. Hence, the main objective of this study was to assess the maternal and neonatal complications of repeat cesarean sections, which is essential for health intervention programs.

## 2. Materials and Methods

### 2.1. Design Setting and Participants

This retrospective study reviewed the electronic records of all women who underwent CSs from 1 January 2023 to 24 July 2023 at three hospitals in the Jazan region,1 of the 13 regions in Saudi Arabia. It is located directly north of Yemen’s border in the southwest region of the kingdom. The Jazan region mainly consists of three cities: Jazan City, Abu Arish, and Sabya. We involved the three general hospitals in these towns to represent the population in the region. The study population was pregnant ladies with one or more previous CSs, who were delivered by repeat CS, and pregnant ladies who were delivered by CS for the first time. The study included those who were delivered by CS electively or if they came for an emergency delivery. All women who met the study criteria were included and separated into two groups for the study: Group I who had a history of fewer than two cesarean sections (undergoing the first or second CS), while Group II had more than two CSs, indicating repeat CSs. These groups were then compared based on obstetric, maternal, and clinical characteristics.

The estimation of sample size for this study was based on the sample size statistical formula:*n* = [(z^2^ × p × q)]/d^2^
where *n* is the initial sample size; p is the anticipated population proportion; z is the standardized variable that corresponds to a 95% confidence level; and d is the absolute precision required.

Using this equation and the parameters of prevalence of CS of 20% [7], 95% confidence interval, and error of not more than 5%, the initial sample size was calculated to be 245 women. For practical reasons, the sample size was increased by 10%, resulting in a final sample size of 270 pregnant women. The final sample size was divided equally among the three selected hospitals.

### 2.2. Data Collection, Study Variables, and Their Definitions

In a predesigned data extraction sheet, the following data were collected from the patients’ medical charts from the three selected hospitals: demographic information including maternal age, residence, education, occupation, and monthly income; medical and obstetric information, including antenatal care, no previous CS and indications, information on the current pregnancy, any history of abdominal pain or vaginal bleeding, any hospitalization before the operations, and the past medical history; information about the last CS, including the type of operation (elective or emergency), its indication, time of the start of uterine contractions and the start of the operation, type of anesthesia, type of abdominal incision, findings during the operation such as fenestration, rectal muscle diastasis, adhesions, site of the urinary bladder, site of the placenta, intrapartum bleeding, estimated blood loss, any injury during operation, any need for tubal ligation, need for hysterectomy, any severe bleeding, any uterine rupture, operation time, incision to delivery time, days of postoperative care, any postoperative complications like bleeding, thromboembolism, sepsis, endometritis, UTI, fever, and wound dehiscence; HB on admission; postoperative HB; and need for blood transfusion. Finally, information about the newborn was recorded: sex, weight, Apgar score at 1 min and 5 min, need for resuscitation or NICU admission, and time of starting breastfeeding.

### 2.3. Data Analysis

The collected data were checked regularly and analyzed using the SPSS (Statistical Package for Social Sciences) software package v26. Descriptive statistics was used to summarize the study variables (e.g., postnatal outcomes and complications, background characteristics) using frequencies and percentages for qualitative variables and means and standard deviations for quantitative variables. Another level of data analysis, including Chi-Squared/Fisher exact tests and logistic regression, was used to test some associations between factors associated with complications of women with multiple cesarean sections. Bivariate logistic regression was used to determine which factors are associated with repeat cesarean sections in the Jazan region. Odd ratios with their 95% CI were reported. A *p* value less than 0.05 was regarded as statistically significant.

## 3. Results

Table 1 presents the patients’ sociodemographic and obstetrical characteristics. Out of all the mothers, the majority (54.5%) were between the ages of 20 and 30 years old. Small percentages (7%) of mothers were in the age group of 41 to 50 years old. More than half (51.9%) of the women were from Abuarish. Nearly all (91.8%) of the mothers received regular antenatal care, with half (50%) being first-time mothers. Only a few (12.7%) of the study participants had chronic health issues, with only 5.2% having diabetes. The sex ratio at birth showed that more than half (53.0%) of the newborns were female.

As shown in Figure 1, most of the CSs were indicated due to failure of progress, followed by breech presentation. However, in Figure 2, approximately a third of all participants had a cesarean section due to a previous cesarean section; this was followed by a breech presentation, preeclampsia, and failure to progress as other frequent indications.

Table 2 shows the intra-operative and postoperative complications associated with CSs among the study groups. About 60% of the operations were due to emergencies compared to only 39.6% as planned elective operations. Spinal anesthesia constituted the most frequently used type (71.3%), followed by general anesthesia and epidural. The most frequent intra-operative maternal complications were intra-peritoneal adhesions (7.5%) and fused abdominal wall layers (7.1%). A low placental site was found in 80 (29.9%) women, 58 (29.7%) in Group I and 22 (30.1%) in Group II.

Furthermore, the most common postoperative complication was requiring a blood transfusion in 59 (22%) women, 46 (23.65) in Group I and 13 (17.8%) in Group II. There were few cases of postoperative infections and wound dehiscence, which occurred in 8 (3%) cases, 7 (3.6%) in Group I and 1 (1.4%) in Group II. One case (0.4%) of paralytic ileus and one case (0.4%) of thromboembolism were reported in Group I.

On the other hand, the reported neonatal complications were a low Apgar score (<5 in the first minute) in 19% of operations, with 2.6% needing resuscitation. Also, due to cesarean delivery, 3.7% of mothers failed to initiate breastfeeding in the first 24 h. There was a significant difference (*p* < 0.001) between Group I and Group II in the type of operation, as an emergency section constituted more than two-thirds of the Group I operations. Furthermore, there was a significantly lower Apgar score (*p* = 0.001) in Group I.

As shown in Table 3, the mean parity of all participants was 1.9 ± 1.1, and the birth weight was 3 ± 0.26 between the delivered neonates. The mean operating time was 48.5 min, which was significantly higher for Group I patients (*p* < 0.001) compared to Group II, where most mothers performed more than two previous cesarean sections. There was no significant change in hemoglobin level between admission and after the CS in the participants; however, there was a need for blood transfusion (about 730 ± 315) among the 60 participants. The postoperative hospital stays were reported to be an average of 3.31 ± 1.5 days without significant differences between Groups I and II.

Table 4 illustrates the factors associated with repeat cesarean sections based on the logistic regression model. The age group was a factor significantly associated with the repeat cesarean sections as those who were in the age group 31–40 [(COR = 2.99, 95% CI: 1.77–5.05, *p* < 0.001)] were more likely to have repeat cesarean sections. In addition, women residing in Jazan and with a poor medical history were more likely to have repeat cesarean sections [(COR = 0.15, 95% CI: 0.08–0.29, *p* < 0.001) and (COR = 29.94, 95% CI: 11.40–78.64 *p* < 0.001), respectively].

## 4. Discussion

While CSs are considered a safe procedure, each subsequent surgery carries additional risks to the mother due to scar tissue formation and potential damage to the surrounding organs. Additionally, CSs could carry risk factors that affect the neonates, such as the need for resuscitation due to respiratory distress and delay in breastfeeding initiation. This analysis evaluated the maternal and neonatal complications of repeat cesarean sections in Saudi mothers in the Jazan region.

In this study, although 60% of the CSs were due to emergencies, the indication was frequently because of prior cesarean sections in nearly half of the mothers; this was followed by a breech presentation, preeclampsia, and failure to progress. Our study results agreed with the previous study conducted in Riyadh city, Saudi Arabia, which reported that 67% of CSs were due to emergencies, with difficult labor, fetal distress, and breech presentation being the most frequent indications. In comparison, in the elective CSs, 33% were indicated due to previous CSs, a breech presentation, and the mother’s requests [10]. Compared to the Arab context, in an Egyptian study, the factors determining CSs are the previous scar in about 50% of cases and fetal distress in 10%; however, a large proportion of the sample did not have any other accompanying indications [11]. Another study conducted in Iraq ranked the top three indications for cesarean sections as having a previous CS, cephalo-pelvic disproportion, and the mother’s request [12]. Elective cesarean sections are known to have better maternal and neonatal outcomes than emergency sections because the former is performed in controlled and planned settings [1,13]. In the current study, the higher rates of emergency CSs in Group I compared to Group II make it difficult to make clear conclusions, as this may lead to discrepancies in maternal and neonatal outcomes. Therefore, to obtain an unbiased result, a larger data set with equitable distribution of emergency and cesarean sections should be considered.

Regarding the intra-operative and postoperative complications associated with CSs among the study groups, the most frequent intra-operative maternal complications of CSs were intra-peritoneal adhesions and fused abdominal wall layers. An attached urinary bladder to the anterior abdominal wall, intrapartum hemorrhage, fenestration, and rectal muscle diastasis were insignificant. Moreover, among the sample population, few women needed tubal ligation. These results agree with a retrospective study that assessed the risks of repeat CSs, which was performed using the hospital records at the Obstetrics and Gynecology Department of Tepecik Training and Research Hospital in Izmir, Turkey, between January 2013 and January 2016. The results showed that although repeat CSs were associated with more adhesions, there were no significant differences in serious morbidities [14].

In comparison, the postoperative complications of requiring a blood transfusion, wound dehiscence, and postoperative infections were reported in fewer cases in Group II. However, paralytic ileus and thromboembolism were reported less in Group I. In the same Saudi study mentioned previously, blood transfusion, ICU admission, HELLP syndrome (with raised liver enzymes and a lower platelet count), and hysterectomy were the most frequent adverse maternal complications [10]. It has been ascertained that emergency CSs are associated with considerable maternal and fatal complications compared to elective CSs [15,16]. The current study revealed that out of 268 cesarean sections, 60% were emergency CSs with 70% performed in Group I who had a history of less than two sections. Comparing the two groups in this study, Group I reported more maternal complications than Group II. This result is the opposite to that of a previous study indicating that maternal complications usually increase in subsequent CS deliveries.

On the other hand, the reported neonatal complications were a low Apgar score (<5 in the first minute) in 19% of the delivered babies which was significantly higher in Group I, with 2.6% of babies needing neonatal resuscitation, and only 3% of the delivered babies were admitted to NICU. Compared to women who delivered vaginally, there was an increased NICU admission rate between babies delivered by CS, which was twice as high as the usual admissions [17]. Furthermore, pulmonary conditions frequently associated with CSs; these include respiratory distress syndrome and transient tachypnea of the newborn, which can result in inefficient expulsion of fetal lung fluid after delivery, impaired gas exchange, respiratory distress, and tachypnea [18]. Additionally, difficulties initiating breastfeeding occurred more frequently in babies delivered through CS, which may be attributed to the mother’s condition after surgery and the baby’s condition due to respiratory distress [19]. A recent systematic review reported that CSs are adversely associated with the initiation of breastfeeding and pointed to the potential association between a mother’s preference for CS and her subsequent decision not to breastfeed [20]. However, a meta-analysis indicated that CSs are not related to breastfeeding initiation if there is satisfactory support for the mother [21]. Fortunately, in this study, only 3.7% of the mothers failed to initiate breastfeeding in the first 24 h, indicating adequate maternal and health professional care about breastfeeding.

The average length of hospital stay is frequently used as a quality measure for medical procedures. For instance, implementing an immediate clinical care pathway lowers the length of hospital stay and treatment costs [22]. Reducing hospital stays following cesarean sections is becoming more prevalent globally, and this reduction in the length of stay following a cesarean section has not been associated with adverse maternal health outcomes [23]. In this study, the length of postoperative hospital stays was 3.31 ± 1.5 days without significant differences between Groups I and II, and this reflects a good indicator of optimal and immediate clinical care that prevents complications that necessitate extended hospital stays.

The current study identified determinants of repeat cesarean sections, including the mother’s age (31–40 years), residence (Jazan), and poor medical history, which were found to be more related to the repeat CSs rather than other variables. These results are consistent with previous studies correlating maternal age and obstetrical history to frequent CS deliveries [24]. In this study, although 5.2% of the mothers had a history of diabetes mellitus and 3% had hypertension, there were no significant associations between these chronic illnesses and repeat CSs.

This study aimed to identify the maternal and neonatal complications of repeat CSs, which is crucial to building a fundamental data set in southern Saudi Arabia. However, certain limitations were encountered during the collection of data. First, although the included hospitals had good recording systems, vital information was missed such as education, occupation, monthly income, and whether the mother had undergone trial of labor after cesarean section. Secondly, the tertiary hospitals in the area were not included, excluding the high-risk women who had undergone more than five previous CSs. Hence, we recommend a future study including more hospitals, specifically the available tertiary hospitals in the region. Moreover, comparing multiple CSs, trial of labor after cesarean, and vaginal birth after CS (VBAC) in future studies is mandatory to evaluate the maternal and neonatal outcomes.

## 5. Conclusions

The most typical complications in this study were intra-peritoneal adhesions and fused abdominal wall layers, which did not increase maternal morbidity. In addition, blood transfusion and postoperative infections were overcome by the availability of antimicrobials and improved blood banking techniques. However, the frequent neonatal complications were a low Apgar score, needing neonatal resuscitation, and intensive care admission. Therefore, repeat CSs remain a safe obstetric procedure with good maternal and fetal outcomes.

## Figures and Tables

**Figure 1 healthcare-11-02799-f001:**
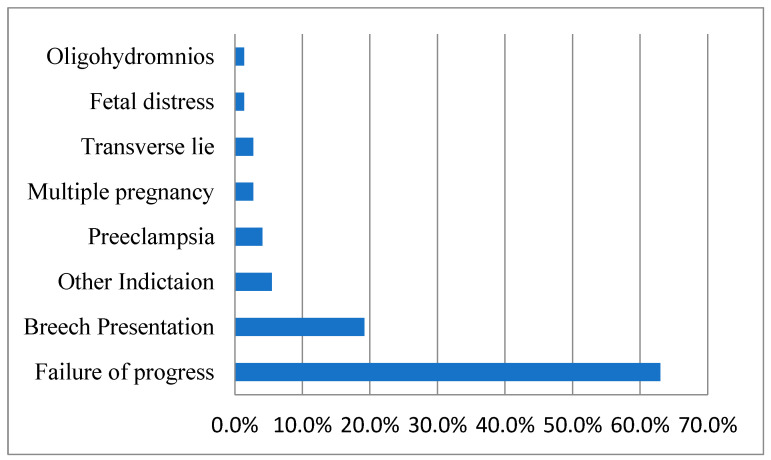
Distribution of the first cesarean section indications under common categories for Group I.

**Figure 2 healthcare-11-02799-f002:**
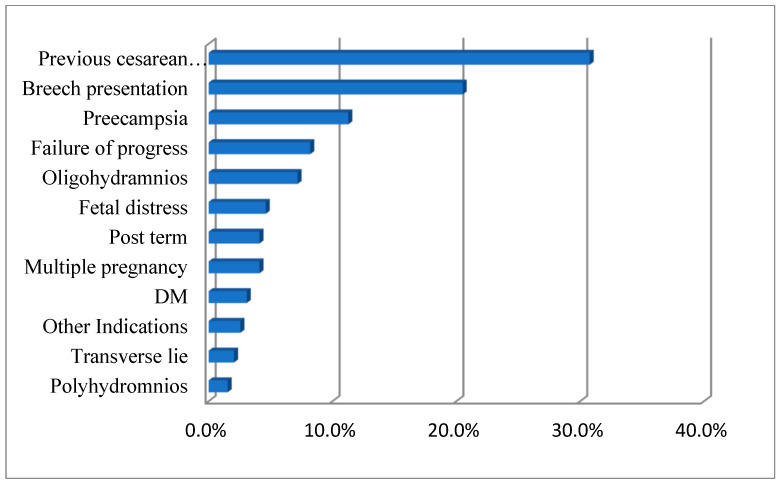
Distribution of current cesarean section indications under common categories for Group II.

**Table 1 healthcare-11-02799-t001:** Maternal demographic and obstetrical characteristics (*n* = 268).

Characteristic		Number	Percentage
Age group (years)	20–30	146	54.5%
	31–40	103	38.4%
	41–50	19	7.1%
Residence	Abuarish	139	51.9%
	Jazan	88	32.8%
	Sabya	41	15.3%
Parity	1	133	49.6%
	2	62	23.1%
	3	49	18.3%
	4	15	5.6%
	5	9	3.4%
Had regular antenatal care	Yes	246	91.8%
	No	22	8.2%
History of any chronic conditions	Yes	34	12.7%
	No	234	87.3%
History of DM	Yes	14	5.2%
	No	254	94.8%
History of HTN	Yes	8	3.0%
	No	260	97.0%
Number of previous CSs	0	133	49.6%
	1	62	23.1%
	2	49	18.3%
	3	15	5.6%
	4	9	3.4%
Need of antenatal admission before operation	Yes	24	9.0%
	No	244	91.0%
Child gender	Female	142	53.0%
	Male	126	47.0%

DM: diabetes mellitus; HTN: hypertension; CS: cesarean section.

**Table 2 healthcare-11-02799-t002:** Intra-operative and postoperative complications associated with CSs among the study groups.

Condition		All		Group I		Group II		*p* Value *
		N	%	N	%	N	%	
Fenestration	No	266	(99.3)	194	(99.5)	72	(98.6)	0.471
	Yes	2	(0.7)	1	(0.5)	1	(1.4)	
Rectal muscle diastasis (separated)	No	267	(99.6)	195	(100.0)	72	(98.6)	0.272
	Yes	1	(0.4)	0	(0.0)	1	(1.4)	
Fused abdominal wall layers	No	249	(92.9)	184	(94.4)	65	(89.0)	0.109
	Yes	19	(7.1)	11	(5.6)	8	(11.0)	
Urinary bladder attached high in the anterior abdominal wall	No	263	(98.1)	193	(99.0)	70	(95.9)	0.126
	Yes	5	(1.9)	2	(1.0)	3	(4.1)	
Intra peritoneal adhesions	No	248	(92.5)	184	(94.4)	64	(87.7)	0.060
	Yes	20	(7.5)	11	(5.6)	9	(12.3)	
Intrapartum hemorrhage	No	263	(98.1)	192	(98.5)	71	(97.3)	0.415
	Yes	5	(1.9)	3	(1.5)	2	(2.7)	
Need for tubal ligation	No	265	(98.9)	194	(99.5)	71	(97.3)	0.180
	Yes	3	(1.1)	1	(0.5)	2	(2.7)	
Early mobilization within 24 h	No	5	(1.9)	5	(2.6)	0	(0.0)	0.201
	Yes	263	(98.1)	190	(97.4)	73	(100.0)	
Postoperative complications	No	267	(99.6)	195	(100.0)	72	(98.6)	0.272
	Yes	1	(0.4)	0	(0.0)	1	(1.4)	
Thromboembolism	No	267	(99.6)	194	(99.5)	73	(100.0)	0.728
	Yes	1	(0.4)	1	(0.5)	0	(0.0)	
Sepsis	No	265	(98.9)	194	(99.5)	71	(97.3)	0.181
	Yes	3	(1.1)	1	(0.5)	2	(2.7)	
Paralytic ileus	No	267	(99.6)	194	(99.5)	73	(100.0)	0.728
	Yes	1	(0.4)	1	(0.5)	0	(0.0)	
UTI	No	260	(97.0)	188	(96.4)	72	(98.6)	0.130
	Yes	8	(3.0)	7	(3.6)	1	(1.4)	
Fever	No	256	(95.5)	185	(94.9)	71	(97.3)	0.319
	Yes	8	(3.0)	7	(3.6)	1	(1.4)	
Need for blood transfusion	No	209	(78.0)	149	(76.4)	60	(82.2)	0.198
	Yes	59	(22.0)	46	(23.6)	13	(17.8)	
Start breastfeeding in the first 24 h	No	10	(3.7)	7	(3.6)	3	(4.1)	0.543
	Yes	258	(96.3)	188	(96.4)	70	(95.9)	
Needed resuscitation	No	261	(97.4)	192	(98.5)	69	(94.5)	0.090
	Yes	7	(2.6)	3	(1.5)	4	(5.5)	
Needed NICU admission	No	260	(97.0)	189	(96.9)	71	(97.3)	0.623
	Yes	8	(3.0)	6	(3.1)	2	(2.7)	
Type of abdominal incision	Midline	19	(7.1)	13	(6.7)	6	(8.2)	0.418
	Pfannenstiel	249	(92.9)	182	(93.3)	67	(91.8)	
Placental site	Lower	80	(29.9)	58	(29.7)	22	(30.1)	0.531
	Upper	188	(70.1)	137	(70.3)	51	(69.9)	
Type of operation	Elective	106	(39.6)	59	(30.3)	47	(64.4)	<0.001
	Emergency	162	(60.4)	136	(69.7)	26	(35.6)	
Type of anesthesia	Epidural	3	(1.1)	2	(1.0)	1	(1.4)	0.152
	GA	74	(27.6)	48	(24.6)	26	(35.6)	
	Spinal	191	(71.3)	145	(74.4)	46	(63.0)	
Apgar score	Normal (10)	217	(81.0)	149	(76.4)	68	(93.2)	0.001
	Low(5)	51	(19.0)	46	(23.6)	5	(6.8)	

UTI: urinary tract infection; NICU: Neonatal Intensive Care Unit; GA: general anesthesia; * *p* value is significant (<0.05).

**Table 3 healthcare-11-02799-t003:** Maternal and surgical outcomes associated with CSs for the two groups.

Factor	All			Group I			Group II			*p* Value
	N	Mean	SD	N	Mean	SD	N	Mean	SD	
Operating time (minutes)	268	48.54	8.95	195	49.74	8.85	73	45.36	8.48	<0.001
Postoperative hospital stay(days)	268	3.31	1.50	195	3.28	1.43	73	3.40	1.66	0.559
Uterine incision to delivery time	268	17.02	4.95	195	17.10	4.71	73	16.81	5.58	0671
Hemoglobin on admission in g/L	182	10.67	1.32	123	10.57	1.40	59	10.87	1.10	0.150
Postoperative hemoglobin in g/L	265	10.19	1.28	193	10.15	1.32	72	10.32	1.19	0.351
Blood loss in mL	230	0.97	0.93	169	1.09	1.01	61	0.66	0.53	0.002
Blood units needed in mL	60	730.00	315.32	48	704.17	327.44	12	833.33	246.18	0.207
Birth weight in kg	268	3.03	0.26	195	3.02	0.25	73	3.03	0.29	0.923
Parity	268	1.9	1.10	195	1.32	0.47	73	3.45	0.71	<0.001

SD: standard deviation; *p* value is significant (<0.05).

**Table 4 healthcare-11-02799-t004:** Bivariate logistic regression to identify factors associated with repeat cesarean sections in Jazan.

Variable		COR	95% CI		*p* Value
			Lower	Upper	
Age group (years)	20–30 (Ref)	1			
	31–40	2.99	1.77	5.05	<0.001
	41–50	2.76	1.02	7.41	0.045
Residence	Abuarish (Ref)	1			
	Jazan	0.15	0.08	0.29	<0.001
	Sabya	1.08	0.52	2.25	0.830
Had regular antenatal care	No (Ref)	1			
	Yes	1.81	0.73	4.46	1.81
History of any chronic condition	No (Ref)	1			
	Yes	1.33	0.65	2.75	0.436
Poor medical history	No (Ref)	1			
	Yes	29.94	11.40	78.64	<0.001

Abbreviations: Ref = reference; COR = crude odds ratio; CI = confidence interval; *p* value is significant (<0.05).

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
