# Peer review of "Multiple Cesarean Section Outcomes and Complications: A Retrospective Study in Jazan, Saudi Arabia"

_healthcare, 2023, doi:10.3390/healthcare11202799_

Round 1
Reviewer 1 Report
In their paper entitled “Multiple Caesarean Sections, Outcomes and Complications: A Retrospective Study in Jazan, Saudi Arabia”, the authors were intended to analysis the outcomes from multiple caesarean sections. I would like to raise the following concerns:
1-The MOH data in Introduction is too old to provide a recent background on CS in this country
2-How to determine three hospitals randomly chosen can be representative of all regions, this can be bias, and experience of the surgeon especially for repeat CS should be factors influencing the results of the study.
3- Authors included emergency CS which obviously increases the occurrence of various complications,how to explore this effect on outcomes.
4-Although the authors calculated the sample demanded, the overall enrolled population had fewer pregnancy comorbidities, and whether it was compared with previous published articles to consider the relationship between the population characteristics and outcomes.
5-There should be a corresponding abbreviation narrative below each table
6-line173-174 font size uniform.
7-Multiple CSs causing an increase in adverse outcomes seems to be an obvious situation, and what the purpose of analysing multiple CSs in the region? is to reduce CS rates? Authors should compare differences with VABC, and the study seemed old-fashioned and with poor value.
Minor editing of English language required.
Author Response
We appreciate your great effort and valuable comments on the manuscript, which we believe have contributed to the development and improvement of the article.

Reviewer 2 Report
Multiple CS: Outcomes and complications
Thank you for giving me the opportunity to read this manuscript.
Overall, this manuscript is concise and well-described following the correct way of statistical analysis.
However, the following points I would like to address for authors to consider to revise.
Abstract: L21 and evaluate its primary outcome~ sounds too vague. The authors need to clarify on it.
Introduction part.
L37-42: First paragraph: The authors need to insert some references for supporting the complications or risks of repeated CS.
L56-47: As for the situation in the Saudi Arabia, the description regarding CS rate there is not updated (2003 is too old), and unclear (265%?). Please state the current situation of CS rate in national or relevant data (MICS, or DHS).
Methods:
L81-82: Group 1 is the first CS (primiparas) and Group 2 is more than second CS. Is my understanding correct?
Please describe geographical characteristics of the region, and the general hospitals (how many deliveries a year, and how much rate of CSs, the number of obstetricians, and midwives I possible).
Figure 1: I would recommend the authors to make two figures od CS in Group 1 and group 2.
Tables 2 and 3: Formatting is required. I cannot see where each p-value goes to which variable.
Table 2 and Table 3 makes sense since the primiparous women are more likely to have complicated delivery (emergency CS).
Discussion: I would guess that the quality of medical services at the targeted hospital are quite good, and the number of group 2 is quite small, hence the results of some complexity may not be significant.( In addition, if the authors excluded emergency CS cases in group 1, the results might be different.) I would highly appreciate it if the authors could discuss more about the results of this study referring the big data conducted in previous studies.
Author Response
Thank you very much for taking the time to review this manuscript. Please find the detailed responses below and the corresponding revisions/corrections highlighted/in track changes in the re-submitted file.

Round 2
Reviewer 1 Report
Thank you for addressing the items previously raised. The manuscript would be suitable for publication.